# Feedforward growth rate control mitigates gene activation burden

Carlos Barajas[1], Hsin-Ho Huang [1], Jesse Gibson[2], Luis Sandoval[3] & Domitilla Del Vecchio [1] ✉

Heterologous gene activation causes non-physiological burden on cellular resources that cells are unable to adjust to. Here, we introduce a feedforward controller that actuates growth rate upon activation of a gene of interest (GOI) to compensate for such a burden. The controller achieves this by activating a modified SpoT enzyme (SpoTH) with sole hydrolysis activity, which lowers ppGpp level and thus increases growth rate. An inducible RelA+ expression cassette further allows to precisely set the basal level of ppGpp, and thus nominal growth rate, in any bacterial strain. Without the controller, activation of the GOI decreased growth rate by more than 50%. With the controller, we could activate the GOI to the same level without growth rate defect. A cell strain armed with the controller in co-culture enabled persistent population-level activation of a GOI, which could not be achieved by a strain devoid of the controller. The feedforward controller is a tunable, modular, and portable tool that allows dynamic gene activation without growth rate defects for bacterial synthetic biology applications.

In bacterial synthetic genetic circuits, genes work in orchestration to accomplish a variety of functions, from monitoring stress level and releasing drugs in the gut[1–4], to sensing environmental pollutants in soil or water[5–10]. In these circuits, genes become dynamically activated and repressed, depending on the environment and on the circuit's state. When a gene of interest (GOI) is activated, cellular resources that the cell would otherwise devote to growth are used by the GOI's expression. This burden on cellular resources decreases growth rate and leads to physiological changes with poorly predictable outcomes that generally hinder the intended performance of the engineered cell[11–17]. Decreased growth rate upon a GOI activation has especially severe consequences when engineered bacteria are in co-culture with other strains. In fact, co-existence of multiple strains in co-culture is contingent on tightly matching growth rates, wherein small growth rate differences between the strains typically lead to extinction of the slower growing strain[18–20]. Therefore, growth rate reduction of an engineered bacterial strain upon a GOI activation, by leading to this strain's extinction in co-culture, also leads to loss of the population-level expression of the GOI.

To mitigate the consequences of gene expression burden, researchers have devised methods that make synthetic gene expression robust to changes in the availability of cellular resources[21–25]. Complementary approaches have also used orthogonal ribosomes for heterologous expression through synthetic ribosomal RNA (rRNA)[26]. However, none of these tools can control growth rate. The problem of controlling growth rate has been addressed by a feedback controller that senses gene expression burden and reduces the GOI's expression to low levels such that growth rate is not affected[27]. While this approach is ideal to maximize protein yield in batch-production, it is not suitable when the GOI needs to be dynamically activated to a specific and possibly high level as in biosensors and genetic logic gates[28,29]. Here, we introduce a feedforward controller that allows the activation of a GOI to any desired level while keeping growth rate constant. The controller co-expresses SpoTH, a modified version of SpoT with only hydrolysis activity, with the GOI.

When the GOI is activated, SpoTH is also expressed and catalyzes the hydrolysis of ppGpp, thereby de-repressing ribosomal rRNA and increasing ribosome level and growth rate[30–32]. We induce the

[1]Department of Mechanical Engineering, Massachusetts Institute of Technology, Cambridge, MA, USA. [2]Department of Bioengineering, Stanford University, Palo Alto, CA, USA. [3]Department of Chemical Engineering, Massachusetts Institute of Technology, Cambridge, MA, USA. ✉e-mail: ddv@mit.edu

expression of RelA+, a variant of RelA protein that exhibits constitutive ppGpp synthesis activity, to elevate ppGpp level in any strain of interest[33,34]. Therefore, we use RelA+ to modulate basal ppGpp level and thus achieve a desired growth rate. We then control the expression of SpoTH to maintain that desired growth rate as a GOI is activated. The controller achieves a constant growth rate as a GOI is activated in multiple strains, at different nominal growth rates, and also in co-culture.

## Results

### Growth rate actuation via SpoTH in strains with elevated ppGpp levels

During balanced exponential growth, ppGpp is the primary regulator of both rRNA and growth rate[30–32] and, in particular, there is an inverse relationship between basal ppGpp level and both rRNA transcription rate, and growth rate[35–37]. Furthermore, during exponential growth, rRNA production rate is the rate-limiting step in the process of ribosome production[30,38]. The RelA/SpoT Homolog (RSH) proteins are responsible for catalyzing the synthesis and hydrolysis of ppGpp as shown in Fig. 1a[39–41]. In particular, the SpoT enzyme is bifunctional with both synthesis and hydrolysis capabilities, with the latter dominating in exponential growth[42], while the RelA enzyme is monofunctional with sole synthesis activity. To actuate growth rate, we exogenously express a modified version of SpoT (Supplementary note 1) with only hydrolysis activity (SpoTH). Activation of SpoTH catalyzes the hydrolysis of ppGpp, as shown directly in[43]. As a consequence of a decreased ppGpp level, we thus expect an upregulation in growth rate (Fig. 1a and mathematical model in Supplementary notes 2 and 3). The growth rate versus SpoTH expression as ppGpp concentration varies, as predicted by our mathematical model (Supplementary notes 2 and 3), is shown in Fig. 1b. Expression of SpoTH also places a load on the cell via resource (e.g., ribosome) sequestration (dashed flat headed arrow in Fig. 1a from SpoTH to ribosomes). Thus, once a sufficient amount of ppGpp has been removed through SpoTH expression, the burden effects from further SpoTH expression overwhelm the upregulation in growth rate due to the removal of ppGpp. In turn, this leads to a non-monotonic response between SpoTH expression and growth rate as observed in Fig. 1b.

We experimentally characterized the ability of SpoTH expression to actuate growth rate in three different strains carrying mutations in the SpoT gene, resulting in different basal levels of ppGpp. Specifically,

we tested the CF944 (*spoT202* allele), CF945 (*spoT203* allele), and CF946 (*spoT204* allele) strains, where the basal ppGpp levels are lowest for CF944 and highest for CF946[31,35,44,45]. Alongside these strains, we also characterized the growth rate response to SpoTH expression in the wild-type MG1655 strain (WT). The genetic circuit used to express SpoTH in these strains is shown in Fig. 2a, b. In particular, Fig. 2b shows how SpoTH expression affects growth rate. For CF945 and CF946, activation of the SpoTH gene increased growth rate by up to 80% and 60%, respectively (Fig. 2c). For MG1655 and CF944, which have lower basal level of ppGpp, we were unable to positively actuate growth rate as the SpoTH gene was activated. Strain CF945 provides the most relative growth rate actuation and thus it is the strain we proceed with.

For a fixed strain, an additional method to tune basal ppGpp level is via the growth medium composition, specifically through the carbon source[32,38,46–48]. Consequently, we also tested four common carbon sources: glucose, glycerol, fructose, and lactose. The nominal growth rate without SpoTH expression was ~0.35 hr$^{-1}$, ~0.32 hr$^{-1}$, ~0.2 hr$^{-1}$, and ~0.12 hr$^{-1}$ and can be increased by up to ~45%, ~50%, ~85%, and ~75% by expressing SpoTH with glucose, fructose, glycerol, and lactose, respectively (Fig. 2d, e).

These data indicate that there is a tradeoff between nominal growth rate and the relative growth rate increase that can be achieved by SpoTH expression (Fig. 2c, e). This tradeoff occurs because the extent to which growth rate can be increased is directly tied to the amount of ppGpp available to be hydrolyzed. That is, high basal ppGpp, yielding lower basal growth rate, allows for larger growth rate increase upon SpoTH expression (Supplementary Fig. 13 and Supplementary note 3). However, the maximum achievable growth rate when SpoTH is expressed should be lower than that when there is no ppGpp and no SpoTH expression (e.g., MG1655 with no SpoTH expression Fig. 2c) since SpoTH expression places a load on cellular resources (dashed arrow in Fig. 2b) and thus on growth rate. An additional consequence of the load that SpoTH expression places on cellular resources is that after most of the ppGpp has been removed via SpoTH expression, any subsequent SpoTH expression will only serve to decrease growth rate (see Supplementary note 2 and 3 for more details).

### Feedforward control of growth rate in the CF945 strain

The feedforward growth rate controller co-expresses SpoTH with the red fluorescent protein (RFP) GOI (Fig. 3a). We refer to this system as the closed loop (CL) system. The open loop (OL) system is a configuration

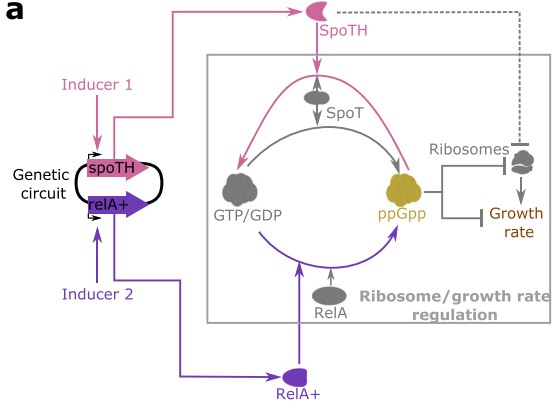

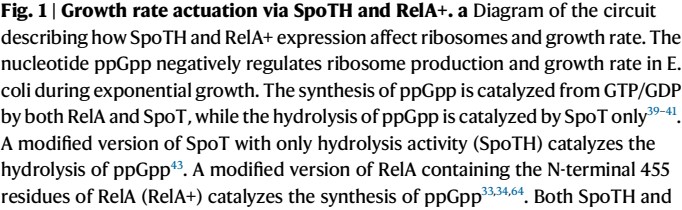

**a**

Genetic circuit

Inducer 1

Inducer 2

SpoTH

SpoT

GTP/GDP

ppGpp

Ribosomes

Growth rate

RelA

Ribosome/growth rate regulation

RelA+

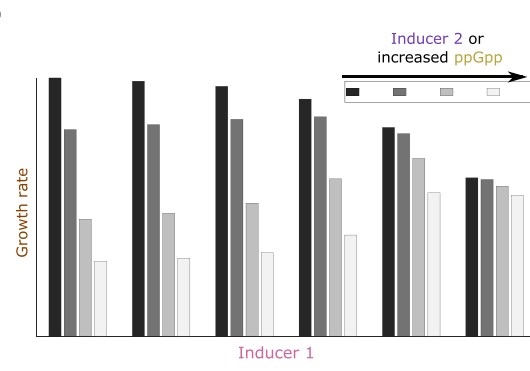

**b**

Growth rate

Inducer 1

Inducer 2 or increased ppGpp

**Fig. 1 | Growth rate actuation via SpoTH and RelA+. a** Diagram of the circuit describing how SpoTH and RelA+ expression affect ribosomes and growth rate. The nucleotide ppGpp negatively regulates ribosome production and growth rate in E. coli during exponential growth. The synthesis of ppGpp is catalyzed from GTP/GDP by both RelA and SpoT, while the hydrolysis of ppGpp is catalyzed by SpoT only[39–41]. A modified version of SpoT with only hydrolysis activity (SpoTH) catalyzes the hydrolysis of ppGpp[43]. A modified version of RelA containing the N-terminal 455 residues of RelA (RelA+) catalyzes the synthesis of ppGpp[33,34,64]. Both SpoTH and

RelA+ are expressed by a synthetic genetic circuit and are under the control of inducible promoters. The dashed flat headed arrow from SpoTH to ribosomes represents the load SpoTH expression places on ribosomes as its mRNA is translated. **b** Growth rate as a function of SpoTH level (inducer 1) for varying ppGpp concentration (different shades of gray). RelA+ (inducer 2) expression increases ppGpp level and thus decreases growth rate. See Supplementary notes 2 and 3 for a mathematical model.

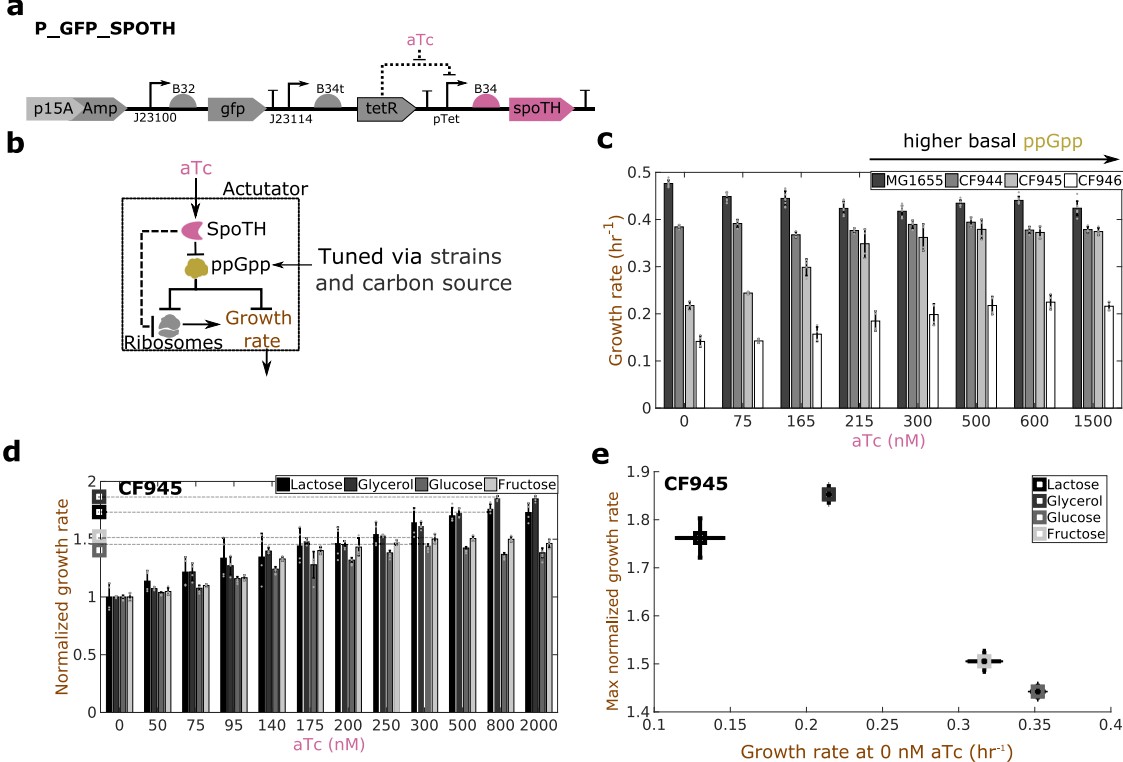

**Fig. 2 | SpoTH gene activation increases growth rate. a** P_GFP_SpoTH plasmid used to activate the SpoTH gene via the inducible pTet promoter. **b** Circuit describing how SpoTH induction via aTc affects ribosomes and growth rate. Addition of aTc increases SpoTH concentration, which lowers ppGpp concentration and consequently upregulates both free ribosome concentration and growth rate[35]. The dashed flat headed arrow from SpoTH to ribosomes represents the load that SpoTH expression places on ribosomes as SpoTH's mRNA is translated. **c** The growth rate as SpoTH is increased in the wild-type MG1655, CF944, CF945, and CF946 strains[35] growing in glycerol as the sole carbon source. **d** Growth rate normalized by the growth rate at aTc = 0 nM, as SpoTH is expressed in CF945 growing

in lactose, glycerol, fructose, or glucose as the sole carbon source. The maximum normalized growth rate for each carbon source is marked by open squares. **e** The maximum normalized growth rate versus the growth rate at aTc = 0 nM for each carbon source. Data are shown as the mean ± one standard deviation ($N = 4$, two biological replicates each with two technical replicates). Individual experimental values are presented as gray dots. The complete experimental protocol is provided in the Materials and Methods section. Plasmid description, plasmid map, and essential DNA sequences are provided in Supplementary Information section *Plasmid maps and DNA sequences*.

where SpoTH is missing (Fig. 3b). Addition of AHL activates the RFP gene, which sequesters cellular resources, including ribosomes, and negatively affects growth rate (upper branch in Fig. 3c)[13,14]. In the CL system, however, addition of AHL also increases SpoTH expression (lower branch in Fig. 3c), which increases ribosome level and growth rate, thereby compensating for the growth rate reduction caused by RFP gene activation. The mathematical model predicts that if the ribosome binding site (RBS) of SpoTH is appropriately tuned, then the availability of ribosomes increases exactly to match the demand for ribosomes by RFP gene activation (Fig. 3d and Supplementary notes 2 and 3). Therefore, we designed four SpoTH RBS's for the CL system with varying strengths (Supplementary note 4).

In fructose, the OL system growth rate drops by over 25% when activating the RFP gene, while for the CL system with RBS 2, the growth rate remains nearly constant when the RFP gene is activated to the same level (Fig. 3e). In glycerol, the OL system growth rate drops by over 45% when activating the RFP gene, while for the CL system with RBS 1, the growth rate drops at most by 10% when we activate the RFP gene to the same level (Fig. 3f). Finally, in lactose, the OL system growth rate drops by over 55% upon RFP gene activation, while for the CL system with RBS 2, the growth rate remains nearly constant for the same RFP gene activation (Fig. 3g). The growth rate versus RFP production rate for other tested CL system's RBS values is shown in Supplementary Fig. 2. From the results of Fig. 2c, e and our mathematical model (Supplementary notes 2 and 3), it follows that the relative growth rate actuation as SpoTH is expressed, is higher for lower nominal growth rates. Therefore, in the feedforward controller design,

we expect that for lower nominal growth rates more substantial growth rate defects from activating the GOI can be canceled, consistent with the results of Fig. 3.

We also considered a control genetic construct, where we replaced SpoTH with a nonfunctional heterologous protein CJB (cjBlue H197S[49]) (Supplementary Fig. 4). This control construct allows to verify that the CL system outperforms the OL system due to the growth rate actuation by SpoTH expression and not because of the configuration change that the RFP mRNA undergoes when RFP is coexpressed with a second gene. This is confirmed since expressing RFP in this control circuit yields even lower growth rates than those of the OL system (Supplementary Fig. 4). This is expected since CJB expression sequesters cellular resources, adding to the burden of activating the RFP gene.

Taken together, these data indicate that the feedforward controller can be easily tuned across different nominal growth rates, which we achieved here by different carbon sources, to ensure no growth rate decrease upon the GOI's activation.

**Feedforward control of ribosomes in common bacterial strain**
To extend the feedforward controller to common bacterial strains, we introduced an inducible RelA+ gene expression cassette to elevate the ppGpp level in any strain of interest (Fig. 4a, b). The *E. coli* RelA+ variant, containing the N-terminal 455 residues of wild-type RelA protein, has constitutive ppGpp-synthesizing activity and its expression has been shown to directly increase ppGpp levels[33,34]. The genetic construct used to express RelA+ and SpoTH is shown in Fig. 4a, b. As expected from the

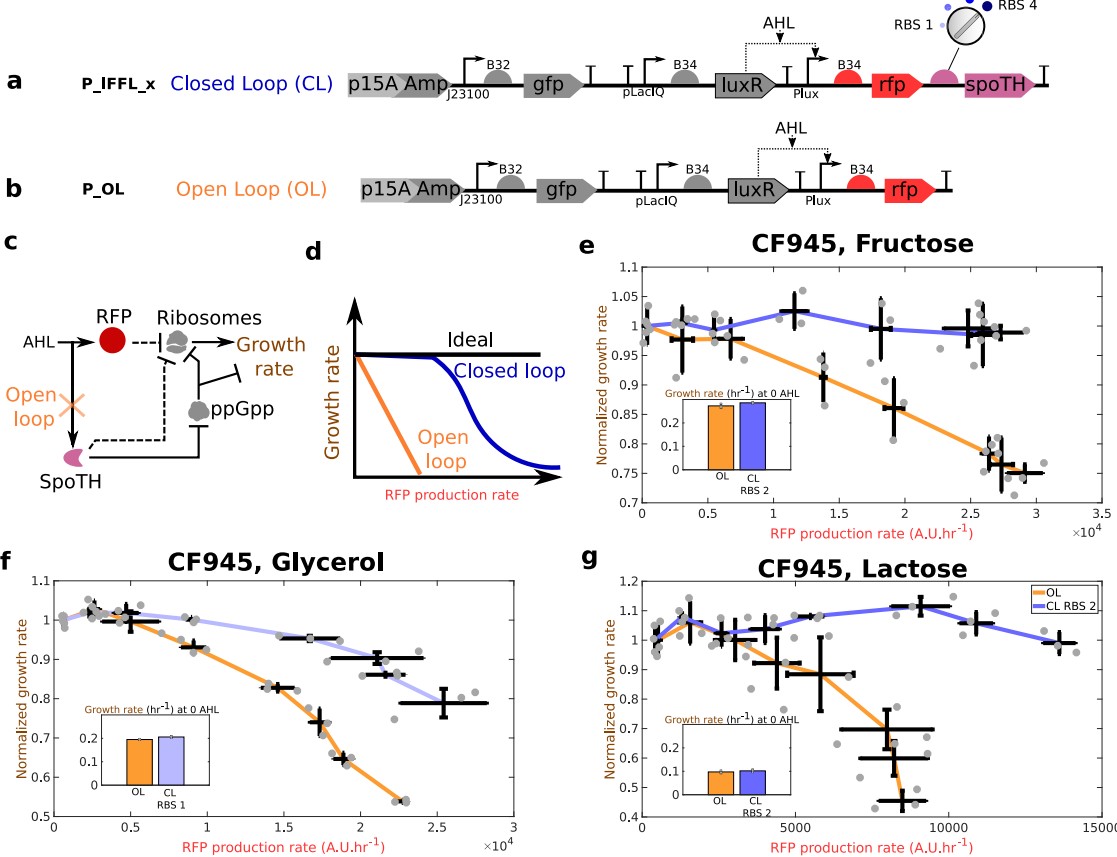

**Fig. 3 | Feedforward ribosome controller compensates for burden caused by a GOI (RFP) activation at different nominal growth rates. a** CL system's genetic construct (P_IFFL_x) co-expresses RFP and SpoTH via the AHL inducible Plux promoter. The SpoTH RBS is used as a tuning parameter (Supplementary note 5). **b** OL system's genetic construct (P_OL) expresses RFP using the AHL inducible Plux promoter. **c** Circuit diagram illustrating how AHL induction affects ribosomes and growth rate for the open loop (OL) or closed loop (CL) systems. In the OL system, SpoTH is not present, so there is only the upper path from AHL to ribosomes. In the CL system, AHL also activates SpoTH expression and hence upregulates ribosome concentration and growth rate. Dashed edges represent sequestration of free ribosomes by a protein's expression. **d** Expected growth rate as the RFP gene is activated for the ideal case, the OL, and the CL systems. **e–g** Growth rate

normalized by the growth rate at 0 nM AHL (nominal growth rate) versus the RFP production rate for the OL and CL systems, using fructose (**e**), glycerol (**f**), and lactose (**g**) as the carbon source. The inset shows the nominal growth rate with no AHL induction. Data for all the RBS's of the CL system tested, AHL induction concentrations used, and GFP production data are shown in Supplementary Fig. 2 and Supplementary Fig. 3. Data are shown as the mean ± one standard deviation ($N = 3$, three biological replicates). Individual experimental values are presented as gray dots. All experiments were performed in the CF945 strain. The complete experimental protocol is provided in the Materials and Methods section. Plasmid description, plasmid map, and essential DNA sequences are provided in Supplementary Information section *Plasmid maps and DNA sequences*.

ability of RelA+ to increase the level of ppGpp, increased levels of RelA+ in MG1655 (WT), TOP10, and NEB strains lead to lower growth rate (Fig. 4c). For a level of RelA+ expression that halves the nominal strain growth rate, SpoTH gene activation upregulates growth rate close to the level with no RelA+ for all three strains (compare growth rate for maximal aTc in Fig. 4d to that for no SAL in Fig. 4c). We conclude that, with constitutive RelA+ expression, SpoTH gene activation allows to increase growth rate in common laboratory strains, thereby enabling transportability of the feedforward controller.

We next evaluated the ability of the feedforward controller to keep growth rate constant as the RFP gene is activated in a TOP10 strain (Fig. 5a, b). To this end, we established three OL systems at different nominal growth rates by transforming the OL system circuit of Fig. 3a in CF944, CF945, and CF946. We then evaluated three genetically identical CL systems all using RBS 2 (Fig. 5a), each with nominal growth rate matching that of the corresponding OL system, which we obtained by adjusting the RelA+ expression level (insets of Fig. 5c–e). This way, both OL and CL systems have matching growth rates before the RFP gene is activated.

When the RFP gene is activated, the growth rate of the OL system drops by 20%, 55%, and 40% in the CF944, CF945, and CF946 strains,

respectively (Fig. 5c–e). In contrast, the growth rates of the associated CL systems, only drop by 5%, 7%, and 15%, respectively, when the RFP gene is activated to the same level (Fig. 5c–e). The RBS of the CL system in Fig. 5e, can be further tuned to prevent a growth rate drop as the RFP gene is activated (Supplementary note 6). The growth rate versus RFP production rate for all CL RBS values tested is shown in Supplementary Fig. 6.

Taken together, these data show that RelA+ expression sets the nominal desired growth rate for the CL system in a strain of interest, and that the SpoTH co-activation with the GOI maintains this pre-set nominal growth rate as the GOI is activated.

## Feedforward controller for persistent GOI expression in co-culture

Engineered bacteria that dynamically express a GOI are often deployed in environments where other microbes are already present. Examples include engineered bacteria delivering biotherapeutics in the gut microbiome or acting as biosensors for water contaminants[4,7]. If the activation of the GOI leads to growth rate defects, then environmental faster-growing organisms will overtake the population, leading to loss of the GOI population-level expression[18,20]. This, in turn, hinders the sensing

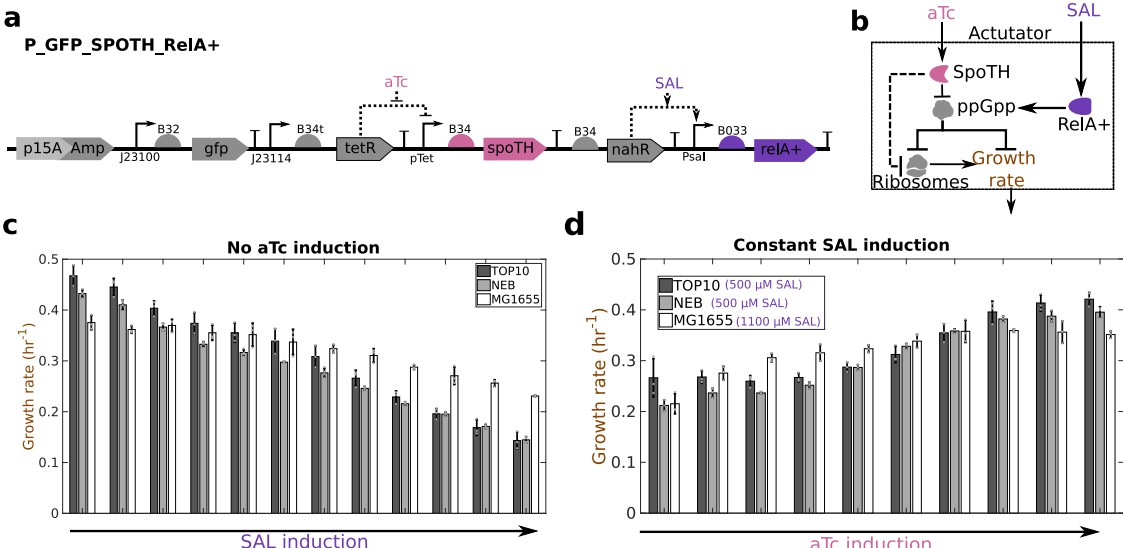

**Fig. 4 | RelA+ expression allows to transport the ribosome controller to a desired bacterial strain. a** P_GFP_SpoTH_RelA+ construct expresses SpoTH via the inducible pTet promoter and RelA+ via the inducible Psal promoter. Plasmid description, plasmid map, and essential DNA sequences are provided in Supplementary Information section *Plasmid maps and DNA sequences*. **b** Circuit diagram depicting the effect of RelA+ induction and SpoTH induction on ribosomes and growth rate. Addition of SAL increases RelA+ concentration and thus upregulates ppGpp concentration[64]. Addition of aTc increases SpoTH concentration, which lowers ppGpp concentration and consequently upregulates both free ribosome concentration and growth rate[35]. The dashed flat headed arrow from SpoTH to ribosomes represents the load SpoTH expression places on ribosomes as its mRNA is translated. **c** Growth rate versus RelA+ induction (SAL) in the TOP10, NEB, and wild-type MG1655, strains growing in glycerol as the sole carbon source. The SAL inductions are [0, 5, 10, 20, 40, 75, 150, 250, 375, 500, 750, 1000] μM for the NEB and TOP10 strains and [0, 10, 20, 30, 50, 100, 175, 250, 375, 500, 750, 1000] μM for MG1655 strain. **d** Growth rate versus SpoTH induction (aTc) for a fixed RelA+ induction in TOP10, NEB, and wild-type MG1655 strains growing in glycerol as the sole carbon source. The aTc inductions are [0, 20, 30, 40, 45, 50, 60, 70, 80, 90] nM for TOP10, [0, 20, 30, 40, 50, 60, 70, 80, 90, 100] nM for NEB, and [0, 40, 80, 120, 160, 200, 240, 280, 320, 360] nM for MG1655. Data are shown as the mean ± one standard deviation (*N* = 3, three biological replicates). Individual experimental values are presented as gray dots. The complete experimental protocol is provided in the Materials and Methods section.

or drug delivery functionality of the engineered cell strain. Similarly, in engineered consortia, where multiple strains are programmed to each accomplish a different but complementary function, the different strains' growth rates should remain sufficiently close to one another despite dynamic activation of genes[17,50,51]. Here, we tackle this problem by employing the feedforward controller to activate the GOI such that the strain's growth rate does not change upon GOI activation.

Specifically, we compare the performance of the OL strain expressing inducible RFP (GOI) to that of the CL strain armed with the feedfroward controller, when co-cultured with a "competitor strain" that constitutively expresses blue fluorescent protein (BFP) (Fig. 6a–c). The performance metric that we use for this comparison is the temporal population-level expression of RFP after its activation, that is, the intensity of RFP normalized by the OD of the co-culture. When grown in isolation and post induction of RFP, the growth rates of the OL and CL strains are initially close to each other and to that of the competitor strain. However, as time progresses, the growth rate of the OL strain drops to about 50% of its original value while that of the CL strain maintains the initial growth rate over time (Fig. 6d).

As a consequence, when OL and competitor strains are in co-culture and the GOI is activated, the population-level intensity of BFP increases, while that of RFP ultimately decreases (Fig. 6e). This dynamic change in the population-level intensity of RFP and BFP can be attributed to the competitor strain overtaking the population due to its higher growth rate (compare blue line to dark gray bars in Fig. 6d). To further verify that this population-level dynamic change was not due to a dynamic change in the expression level of BFP and RFP, we tracked the same biological replicates as in Fig. 6 in monoculture, which showed constant BFP and RFP intensity throughout the time course (Supplementary Fig. 8). When the CL and competitor strains are in co-culture and the GOI is activated, the population-level intensity of BFP and RFP settle to a constant level (Fig. 6f), consistent with the adaptation of the

growth rate of the CL strain to its initial value post induction of the GOI (Fig. 6d, light gray bars). Therefore, we conclude that the CL strain, by preventing a steady decrease in growth rate upon activation of the GOI, also allows persistent GOI population-level expression.

## Discussion

The alarmone ppGpp has been referred to as the "CEO of the cell", whose job is to optimally regulate resources for growth based on environmental conditions and current translational activity[52]. In this paper, we exploited the inverse relationship between ppGpp level and growth rate[35–38] to engineer an actuator that upregulates growth rate (Fig. 1). Specifically, the actuator exogenously expresses a modified version of SpoT with only hydrolysis activity (SpoTH) and, in strains with elevated basal ppGpp level, activation of the SpoTH gene upregulates growth rate (Fig. 2). We demonstrated growth rate actuation first in strains with elevated basal ppGpp level and by tuning the carbon source in the growth media (Fig. 2). Other methods such as tuning the amino acid concentration in the media could also be considered[35]. We then made the actuator portable to common laboratory strains by artificially raising ppGpp's level through the expression of the RelA+ enzyme (Figs. 4 and 5).

We employed the actuator to create a feedforward controller of growth rate that compensates for the burden on cellular resources observed in the form of growth rate defects due to activating a GOI (Figs. 3 and 5). The controller co-expresses SpoTH with the GOI (RFP); therefore, when the GOI is activated, SpoTH is also activated, which increases growth rate. This increase in growth rate, when the SpoTH RBS is well tuned, exactly compensates for the growth defects due to the GOI's activation, leading to no change in growth rate (Fig. 3). The feedforward controller can be implemented for any GOI by co-expressing SpoTH with it and by tuning the SpoTH RBS based on the GOI's load on the cell resources. The design is also tunable and modular.

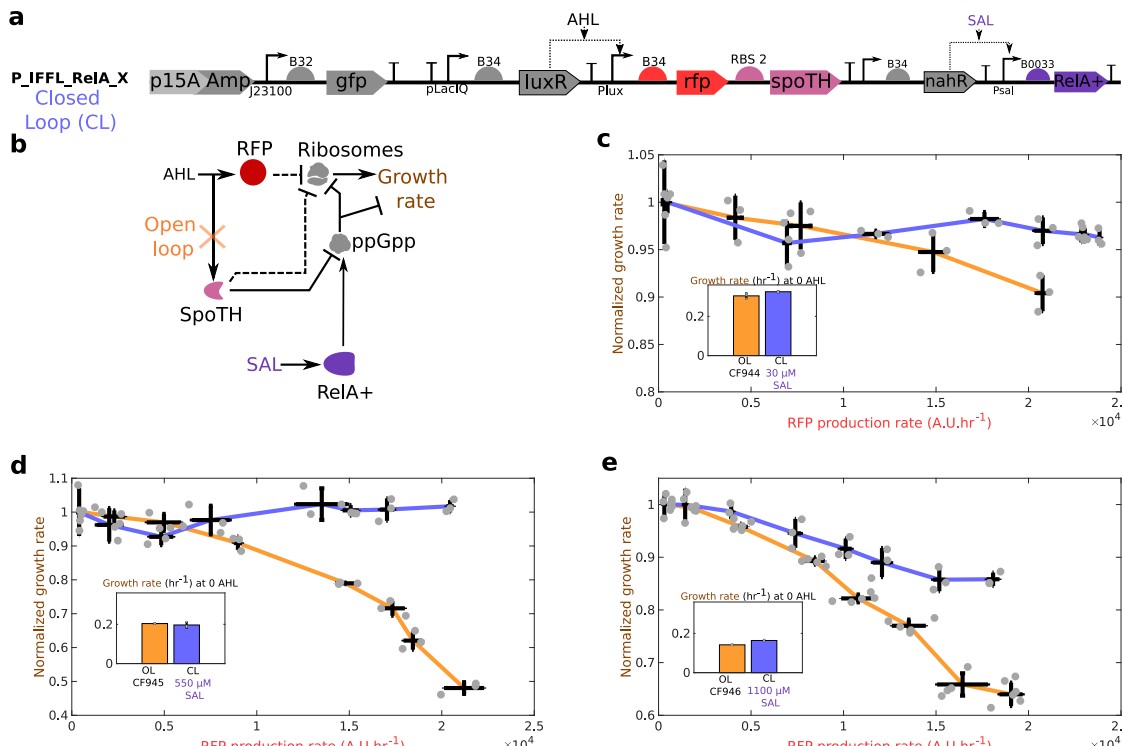

**Fig. 5 | Feedforward ribosome controller compensates for burden caused by activation of the RFP gene in a common laboratory strain and across different nominal growth rates. a** CL system's genetic construct (P_IFFL_RelA_2) co-expresses RFP and SpoTH via the AHL inducible Plux promoter. The SpoTH RBS is fixed to RBS 2 (Supplementary note 5). RelA+ is expressed using the SAL inducible Psal promoter. **b** Circuit diagram depicting the effect of activating RFP (AHL input) on ribosomes and growth rate for the open loop (OL) or closed loop (CL) systems. In the OL system, SpoTH is not present, so there is only the upper path from AHL to growth rate. In the CL system, AHL also activates SpoTH production and hence upregulates ribosome concentration and growth rate. Dashed edges represent sequestration of free ribosomes by a protein's expression. RelA+ activation via SAL sets the basal level of ppGpp and thus the nominal growth rate[64]. **c**–**e** Growth rate

normalized by the growth rate at 0 nM AHL (nominal growth rate shown in the inset) versus the RFP production rate for the OL in CF944 (**c**), CF945, (**d**), and CF946 (**e**) and CL systems in TOP10. For the CL system, RelA+ expression is set to match the growth rate of the OL strain. The inset shows the nominal growth rate with no AHL induction. Data for all RBS's of the CL system tested, AHL induction concentrations used, and GFP production data are shown in Supplementary Fig. 6 and Supplementary Fig. 7. Data are shown as the mean ± one standard deviation ($N = 3$, three biological replicates). Individual experimental values are presented as gray dots. All experiments were performed with glycerol as the sole carbon source. The complete experimental protocol is provided in the Materials and Methods section. Plasmid description, plasmid map, and essential DNA sequences are provided in Supplementary Information section *Plasmid maps and DNA sequences*.

While the primary focus of this study was to exploit the inverse relationship between ppGpp level and growth rate to create a growth rate controller, there is also an inverse relationship between ppGpp level and ribosome production rate[53]. Given that ribosomes are a key resource in protein translation, we expect that our controller can also actuate protein production rate. The ability to control protein production rates can be leveraged to address the critical issue that the activation of a GOI lowers the protein production rate of other genes in the cell, thus indirectly coupling gene expression[13,14]. To test how SpoTH expression actuates protein production rates, we also measured the protein production rate of a constitutively expressed GFP gene for the experiment associated with Fig. 2c. Specifically, GFP production rate increases for MG1655, CF944, CF945, and CF946, by 22%, 45%, 90%, and 65%, respectively, when SpoTH is expressed (Supplementary Fig. 1). For the experiment associated with Fig. 2d, GFP production rate increases by -55%, -60%, -100%, and -150% when expressing SpoTH with glucose, fructose, glycerol, and lactose as the carbon sources, respectively (Supplementary Fig. 1). For the experiment associated with Fig. 4, we showed that RelA+ expression also actuates GFP production rate (Supplementary Fig. 5).

For conditions associated with low nominal growth rates (e.g., cells growing in lactose), we observed the greatest relative protein production actuation as SpoTH was expressed (Fig. 1). Consequently, for sufficiently low growth rates, the feedforward controller can also be tuned to keep the production rate of any constitutively expressed protein

constant as the GOI is activated. Specifically, for the data associated with Fig. 3 and Fig. 5, for low nominal growth rates, the CL system allows also GFP production rate to stay approximately constant when the RFP gene is activated, where it otherwise drops by over 70% for the OL system (Supplementary Fig. 3 and Supplementary Fig. 5). However, the SpoTH RBS that keeps protein production rate constant as the GOI is expressed is not the same as the one that keeps growth rate constant. A constant protein production rate implies that as the GOI is expressed, the changes in ppGpp due to SpoTH expression render a net zero change in free cellular resources responsible for protein production (e.g., ribosomes). However, this change in ppGpp also directly modulates the concentration of growth-related proteins[54,55]. As a consequence, growth rate can increase while protein production rates decrease as SpoTH is expressed (compare Fig. 2 and Supplementary Fig. 1 and see also mathematical model in Supplementary note 3). In future applications, the feedforward controller may be used synergistically with previously engineered controllers that maintain protein production rate constant once a resource competitor (GOI) is activated, but cannot maintain growth rate constant[21-25]. In fact, the concurrent implementation of the SpoTH feedforward controller with these controllers will maintain both a constant growth rate and constant protein production rate of a protein of interest, when a GOI is activated.

The SpoTH actuator can also mitigate the growth defects caused by activation of a toxic protein, such as dCas9[56,57]. With the SpoTH actuator, we could reach without growth defects a dCas9 production

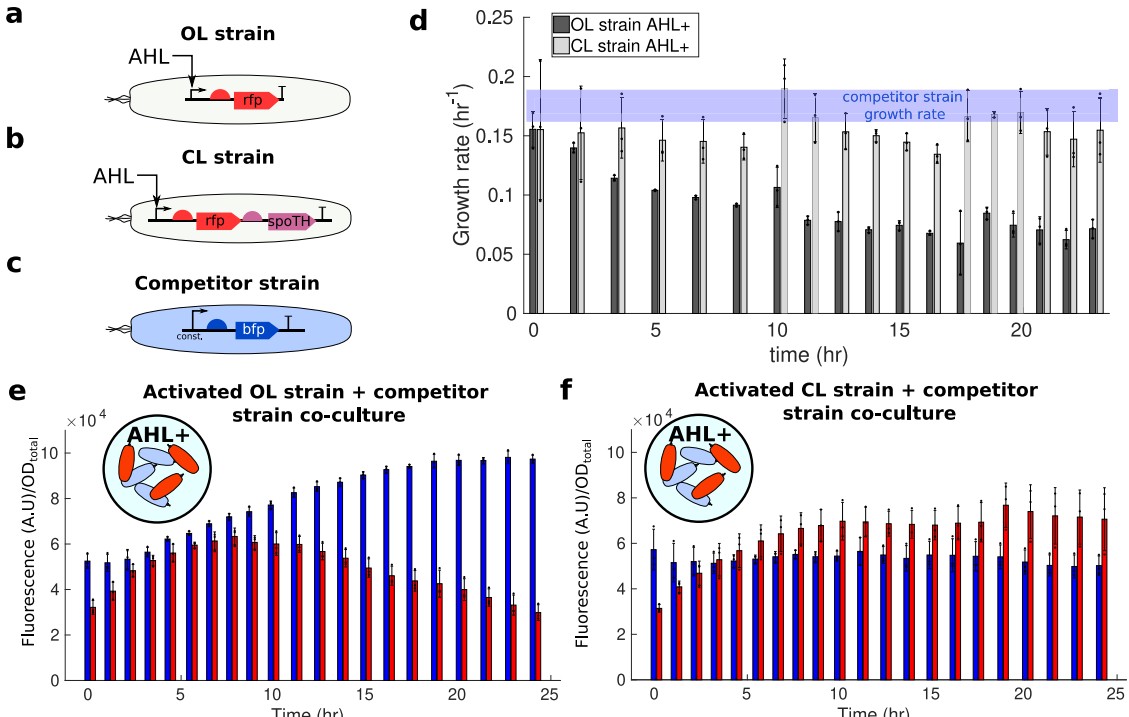

**Fig. 6 | Feedforward controller promotes persistent GOI expression in a co-culture with a competitor strain. a** The OL strain consists of P_OL in CF945. This strain expresses RFP using the AHL inducible Plux promoter. **b** The CL strain consists of P_IFFL_1 in CF945. This strain co-expresses RFP and SpoTH using the AHL inducible Plux promoter. **c** The competitor strain has P_BFP in TOP10. This strain constitutively expresses BFP. **d** Temporal responses of growth rate for the OL and CL strains grown in isolation post activation of the RFP gene (GOI) through AHL induction. The growth rate of the competitor strain grown in isolation is shown with a blue line (see Supplementary Fig. 8 for raw data). **e, f** Temporal responses of RFP (red) and BFP (blue) fluorescence normalized by the total OD of the co-culture

(OD$_{total}$) for the OL and competitor strains co-culture in **e**, and for the CL and competitor strains co-culture in **f**. AHL+ denotes media containing the AHL inducer at 27.5 nM concentration. The growth rate and fluorescence of each strain for all biological replicates was simultaneously tracked in isolation (Supplementary Fig. 8). Data are shown as the mean ± one standard deviation ($N = 3$, three biological replicates). Individual experimental values are presented as black dots. All experiments were performed in media with glycerol as the sole carbon source. The complete experimental protocol is provided in the Materials and Methods section. Plasmid description, plasmid map, and essential DNA sequences are provided in the Supplementary Information section *Plasmid maps and DNA sequences*.

rate that would otherwise cause a 40% decrease in growth rate without SpoTH expression (Supplementary Notes 7). We estimate that this production rate is at least four times higher than that reachable without growth defects without SpoTH expression (Supplementary Notes 7). These results have direct applications to CRISPRi-based genetic circuits where dCas9 should be at high concentrations to minimize the effects of its sequestration by multiple sgRNAs[58–60]. However, dCas9 toxicity limits its concentration to ranges where sequestration effects are prominent[58].

Persistent population-level expression of a GOI in a strain that shares the environment with competing organisms is hampered by growth rate imbalances that follow the GOI activation[4,20]. We applied the feedforward controller to achieve persistent population-level expression of RFP (GOI) in a strain co-cultured with a competitor strain (Fig. 6). In applications, we can use the RelA+ inducible expression cassette to set the strain's nominal growth rate to a desired level chosen to match the growth rate of other competitor strain(s), thereby achieving co-existence. The feedforward controller co-expressing SpoTH with the GOI then guarantees that this desired growth rate does not drop as the GOI is dynamically activated, thereby enabling persistent population-level expression of the GOI. This tool will thus be critical in future multi-strain systems that implement division of labor by running different genetic circuits with distinct, yet complementary, functionalities in the different strains[50,51]. Population controllers have been implemented to promote co-existence in multi-strain systems. However, these controllers require the growth rates of each strain to be sufficiently close to one another[20]. In these systems, GOI activation in one strain may lower growth rate beyond the co-existence limits, at

which point co-existence is lost despite the population controller. Our feedforward controller can be used synergistically with population controllers to ensure co-existence in multi-strain systems when expression of a GOI is dynamically modulated in each of the strains.

The role SpoTH/RelA+ play in upregulating/inhibiting growth rate, may alternatively be achieved by a toxin-antitoxin system[61,62]. An advantage of using direct regulators of ppGpp, such as SpoTH and RelA+, lies in the fact that the effect of these regulators on both ppGpp and growth rate have been characterized extensively and is also constantly evolving[52]. So, as we gain more insight on this pathway, we may uncover opportunities to further optimize the SpoTH actuator. For example, other enzymes like Mesh1[63,64] and SpoT E319Q[65] have been shown to catalyze the hydrolysis of ppGpp and can, in principle, serve in alternative actuator designs in place of SpoTH. Furthermore, this work repurposed tools originally employed in microbiology to study the ppGpp pathway (e.g., SpoTH, RelA+, and CF94x strains) for an engineering application. Therefore this work, more generally, motivates the use of microbiology research for engineering purposes. Finally, given that ppGpp and the RelA/SpoT homologs (RSHs) are universally conserved in bacteria and even appear in eukaryotes, including in humans, our controller may be transported across organism[66]. In particular, the generalization of our controller to other organisms can play a key role when designing multi-organism systems where each member is endowed with growth rates that are robust to gene activation.

The feedforward controller is a tunable, modular, and portable tool that allows dynamic modulation of a GOI's expression to possibly high-levels without substantially affecting growth rate. It will thus be a tool useful for all those applications where engineered bacteria need to

co-exist with environmental species or with other engineered strains, while running circuits in which genes become dynamically activated.

## Methods

### Bacterial strain and growth

The bacterial strain used for genetic circuit construction was E. coli NEB10B (NEB, C3019I) and LB broth Lennox was the growth medium used during construction. Characterization was performed using the CF944, CF495, CF946[35], MG1655 (CGSC, 6300), and TOP10 strains. Characterization experiments were done using M9 minimal medium supplemented with 0.2% casamino acids,1 mM thiamine hydrochloride, ampicillin (100 µg/mL), and either 0.4% glucose, 0.4% fructose, 0.4% glycerol, or 2 g/L lactose (the specific carbon source used for each experiment is specified in the figure caption).

### Microplate photometer protocol

This protocol was used to generate the data in all figures in the main text except that of the co-culture experiment (Fig. 6). Cultures were prepared by streaking cells from a 15% glycerol stock stored at −80 °C onto a LB (Lennox) agar plate containing 100 µg/mL ampicillin and incubated at 37 °C. Isolated colonies were picked and grown in 2 ml of growth medium in culture tubes (VWR, 60818-667) for 12–24 hours at 30 °C and 220 rpm in an orbital shaker. Cultures were then diluted to an OD at 600 nm ($OD_{600\,nm}$) of 0.0075 and grown for an additional 6 hours in culture tubes to ensure exponential growth before induction. Cultures were then induced and plated onto 96-well-plate (Falcon, 351172). The 96-well plate was incubated at 30°C in a Synergy MX (Biotek, Winooski, VT) microplate reader (BioTek Gen 5 v1.11.5 software) in static condition and was shaken at a fast speed for 3 s right before OD and fluorescence measurements. Sampling interval was 5 minutes. Excitation and emission wavelengths to monitor GFP fluorescence are 485 (bandwidth = 20 nm) and 513 nm (bandwidth = 20 nm), respectively, and the Sensitivity = 80. Excitation and emission wavelengths to monitor RFP fluorescence are 584 (bandwidth = 13.5 nm) and 619 nm (bandwidth = 13.5 nm), respectively and the Sensitivity = 100. Sampling continued until bacterial cultures entered the stationary.

### Microplate photometer protocol for co-culture experiment

This protocol was used to generate the data for the co-culture experiment (Fig. 6). Cultures were prepared by streaking cells from a 15 % glycerol stock stored at −80 °C onto an LB (Lennox) agar plate containing 100 µg/mL ampicillin and incubated at 37 °C. Isolated colonies were picked and grown in 2 ml of growth medium in culture tubes (VWR, 60818-667) for 12–24 hours at 30 °C and 220 rpm in an orbital shaker. Cultures were then diluted to an OD at 600 nm ($OD_{600\,nm}$) of 0.0075 for the OL and CL strain and 0.0045 for the competitor strain. After four hours the competitor strain was induced with 550 µM SAL and after 6 hours the OL and CL strains were induced with 27.5 nM AHL. The cultures were then plated onto 96-well-plate (Falcon, 351172) and grown until the optical density was above $OD_{600\,nm} = 0.02$ and were then mixed to bring the optical density of the co-culture to $OD_{600\,nm} = 0.02$. The biological replicate of each culture was simultaneously tracked in isolation (mono-culutre). The cultures were then grown until one of the co-cultures reached $OD_{600\,nm} = 0.2$ and then all cultures were diluted to $OD_{600\,nm} = 0.035$, this dilution process was repeated three times (see Supplementary Fig. 9 for growth curves). The 96-well plate was incubated at 30°C in a Synergy MX (Biotek, Winooski, VT) microplate reader (BioTek Gen 5 v1.11.5 software) in static condition and was shaken at a fast speed for 3 s right before OD and fluorescence measurements. Sampling interval was 5 minutes. Excitation and emission wavelengths to monitor BFP fluorescence are 400 (bandwidth = 9 nm) and 460 nm (bandwidth = 9 nm), respectively and the Sensitivity = 80. Excitation and emission wavelengths to monitor GFP fluorescence are 485 (bandwidth = 9 nm)

and 513 nm (bandwidth = 9 nm), respectively, and the Sensitivity = 80. Excitation and emission wavelengths to monitor RFP fluorescence are 584 (bandwidth = 13.5 nm) and 619 nm (bandwidth = 13.5 nm), respectively, and the Sensitivity = 100.

### Calculating growth rate and protein production rates

The media background OD (0.08 $OD_{600\,nm}$), GFP (100 A.U), and BFP (2800 A.U) were subtracted from the data prior to any calculations. To ensure the data analyzed was coming from cells in exponential growth, only OD values (adjusted for background) of $OD_{600\,nm} = 0.06$ and $OD_{600\,nm} = 0.14$ were considered except for experiments done in lactose where the range was $OD_{600\,nm} = 0.06$ and $OD_{600\,nm} = 0.1$, since cells growing in lactose entered stationary phase at lower OD values.

To dampen noise before differentiating, the data was then filtered using a moving average filter. Given a signal with $n$ measurements $\mathbf{y} = [y_1, y_2, ..., y_{n+1}]$ sampled at a constant period $\Delta t$, we apply the moving average filter as follow:

$$d_j = \sum_{r=-2}^{2} \frac{y_{j+r}}{5}, \quad \forall j \in [3, 4, \ldots, n-1],$$

where $\mathbf{d} = [d_1, d_2, ..., d_{n+1}]$ is our filtered signal with boundary points identical to those of $\mathbf{y}$ ($d_1 = y_1$ and $d_2 = y_2$).

The growth rate is calculated from the filtered OD signal by performing linear regression (in a least-squares sense) on the log of the signal and taking the slope of the fit. The temporal growth rate data from Fig. 6d was calculated by partitioning the OD versus time data (Supplementary Fig. 9) into the time intervals shown in Fig. 6d and then calculating the growth rate of each individual partition per the above method.

The RFP and GFP production rates were calculated in a similar manner as[14]. Denoting $GFP(t_i)$ and $RFP(t_i)$ as the filtered GFP and RFP signal measured by the plate reader at time $t_i$, the GFP production rate ($\alpha_{GFP}(t_i)$) and RFP production rate ($\alpha_{RFP}(t_i)$) are given by

$$\alpha_{GFP}(t_i) = \frac{GFP(t_{i+1}) - GFP(t_{i-1})}{2(t_{i+1} - t_{i-1})OD(t_i)}, \quad \alpha_{RFP}(t_i) = \frac{RFP(t_{i+1}) - RFP(t_{i-1})}{2(t_{i+1} - t_{i-1})OD(t_i)},$$

where $OD(t_i)$ is the filtered OD level.

### Genetic circuit construction

The genetic circuit construction was based on Gibson assembly[67]. DNA fragments to be assembled were amplified by PCR using Phusion High-Fidelity PCR Master Mix with GC Buffer (NEB, M0532S), purified with gel electrophoresis and Zymo clean Gel DNA Recovery Kit (Zymo Research, D4002), quantified with the nanophotometer (Implen, P330), and assembled withGibson assembly protocol using NEBuilder HiFi DNA Assembly Master Mix(NEB, E2621S). Assembled DNA was transformed into competent cells prepared by the CCMB80 buffer (TekNova, C3132). Plasmid DNA was prepared by the plasmid miniprep-classic kit (Zymo Research, D4015). DNA sequencing used Quintarabio DNA basic sequencing service. Primers and gBlocks were obtained from Integrated DNA Technologies. The list of constructs and essential DNA sequences can be found in Supplementary Information section *Plasmid maps and DNA sequences*. The lists of plasmids and primers are provided in Supplementary Data.

### Reporting summary

Further information on research design is available in the Nature Portfolio Reporting Summary linked to this article.

## Data availability

Simulation, fluorescence, and growth rate data generated or analyzed during this study are included in the paper and its Supplementary Information files. A reporting summary for this article is available as a

Supplementary Information file. Source data are provided with this paper.

## Code availability

Custom MATLAB (The MathWorks, Inc., Natick, MA, USA) codes are used to process the experimental data. A Supplementary Software file is provided.

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

## Acknowledgements

This work was supported in part by NSF Expeditions, Grant Number 1521925, NSF RoL Award, Grant Number 1840257, the NSF Graduate Research Fellowships Program, and the Ford Foundation Predoctoral Fellowship. We thank Dr. Cashel, Dr. Potrykus, and Dr. Fernández-Coll for providing the CF944, CF945, and CF946 strains and their helpful discussion on ppGpp.

## Author contributions

D.D.V. and C.B. designed the study; H.H., J.G., L.S., and C.B. designed and built the genetic circuits; C.B. performed the experiments; C.B. analyzed the data; C.B developed the mathematical models; C.B. and D.D.V. wrote the paper.

## Competing interests

The authors declare no competing interests.
