## [Peer Review File · Nature Communications]

Reviewers' Comments:

Reviewer #1:

Remarks to the Author:

In this study, Barajas et al. designed and verified a novel feedforward ribosome control system to mitigate the metabolic burden caused by synthetic gene expression and maintain the constant host cell growth rate. This is important and timely research given that metabolic burden and cell growth rate variation upon synthetic gene expression could significantly limit the design and application of synthetic gene circuits. The feedforward controller they developed used a modified SpoT enzyme (SpoTH), which is inducible upon aTC and only has a hydrolysis activity to lower ppGpp level and thus derepress ribosomes. They tested this controller in multiple strains and growth medium conditions and found that it works very well, especially under conditions with high ppGpp levels and lower growth rates. In addition, they also combined it with an inducible RelA+ cassette to further set the basal level of ppGpp and nominal growth rate. Lastly, they test the controller in the consortia system, which is also very important and widely used in the field but imbalanced growth rates among the strains are a serious problem. Co-culturing with a competitor strain did not affect the average gene expression level in the system with a feedforward controller. Overall, this is a very solid study with rigorous analysis and elegant design and will have an immediate impact on the community. Here are some comments I believe will improve the manuscript.

Major comments:

1. The trade-off between the nominal growth rate and the relative growth rate and the non-monotonic curves shown in Fig.1b is very interesting. However, in the main text, this is not explained in detail. Is this related to the burden caused by the controller itself? Is it because of the saturation level of ppGpp? I believe that this is important information to have in the main text so the reader will understand under what conditions the controller works best intuitively.
2. There seems to be inconsistency between the well-maintained growth rate and a strong decrease in the GFP production rate in Fig. 9 and 13. I understand the authors have some discussion on it. But I suggest the author move the discussion into the main text. I believe this is reasonable if we consider that the resource is also very limited in the system with the controller even if the growth rate is maintained very well.

Minor comments:

1. The authors showed the design of RelA+ in Fig. 1a but did not have any description of it. I suggest that this is either be removed from the diagram and left to Fig. 4 or added some description of the design idea of RelA+ in the beginning.
2. The authors need to explain the direct link from SpoTH to Ribosome in Fig. 2b, in the caption as well as in the main text. Maybe it is a good idea to relate it to the trade-off in the first subsection of Results.
3. It is kind of difficult to map Fig. 2c with Fig. 1b.
4. Given that the nominal growth rate in OL and CL systems is almost the same, is it necessary to normalize the growth rate in Fig. 3 and 5, especially in Fig. 5, where the growth rate matches very well?
5. In the system with RelA+, should the inhibition of Ribosomes mediated by RelA+ also be considered as well? Is there any potential trade-off as well, similar to the SpoTH?
6. Two relevant references (PMID: 33558556, 32251409) are worth to be cited if possible when discussing the effects of decreased growth rate on the performance of engineered cells (page 2, line 25).
7. What is the potential reason for the increase in the growth rate under RFP production in Fig. 8 and 12?

Reviewer #2:

Remarks to the Author:

The paper of Barajas et al. explores regulation of cell growth by modulating the intracellular levels of the alarmone ppGpp. The results show that expression of SpoTH that catalyzes ppGpp decomposition stimulates growth of cells with a sufficiently high basal level of ppGpp. Overexpression of RelA+, a ppGpp synthase, has an opposite effect. By adjusting the relative levels of the two enzymes, or only of one of them, it is possible to tune the growth rate of cells, including those producing the desired protein of interest.

Critique

The general idea of reducing cell growth by producing a toxin and reverting the effect by expressing an antidote is not new. This is the underlying principle of the toxin-antitoxin systems, of coexpressing the antibiotic biosynthetic operons and resistance proteins, of the secondary metabolites and the relevant efflux pumps, and many others. A possible advantage of the SpoTH/RelA+ is a possibility of fine-tuning the growth rate of the cells, even though it is unclear whether SpoTH/RelA+ system is any better in this sense than any other of the aforementioned systems.

The experiments presented in the manuscript are clean and the results are valid. The paper is sufficiently well written even though the clarity of writing could be further improved. However, the interpretations of the results are far from being convincing. In this area, I have several major concerns:

1. The authors presume that expression of SpoTH or RelA+ affects cell growth (one way or the other) by changing the intracellular level of ppGpp. Although this is generally a reasonable assumption, without direct measurements of the ppGpp level, it is nothing but an assumption. Thus, following the authors logic (see below), decreased growth of RelA+ expressing cells could be due to the 'ribosome sequestration' rather than ppGpp accumulation. Therefore, either the levels of ppGpp levels need to be measured directly or any relation to these levels need to be presented as only tentative.

2. II. 95-96. The authors claim that activation of rfp transcription from a plasmid negatively affects growth rate because it "sequesters ribosomes". The same reason is used for explaining why expression of other proteins of interest reduces cell growth. However, expression of many proteins can negatively affect cell for a number of reasons other than ribosome sequestration, while expression of the others may have no effect. Accordingly, what is the evidence that slow cell growth is caused by sequestration of the ribosomes? How large is the fraction of ribosomes that would be 'sequestered' by translating a single gene transcribed from a moderate promoter (Plux) from a low-copy number plasmid (p15A ori)?

3. II. 73-75 and II. 784-797. Measuring ribosome concentration by following GFP expression is a questionable approach. It is based on yet another unsubstantiated assumption that translation elongation rate is not affected by varying ppGpp concentrations. However, given that ppGpp controls transcription of not only rRNA but also tRNA as well as expression of amino acid biosynthetic operons and influx pumps, there are all the reasons to expect that elongation rate would be different in cells with high or low levels of ppGpp. Therefore, without a more direct measurement of ribosome content, it is hard to know how SpoTH expression affects the amount of free ribosomes in the cell.

Minor point:

II.45-46: It is confusing that "to achieve derepression of the rRNA" one activates "ppGpp synthesis activity". The next sentence is even more confusing, not only grammatically, but also its sense is unclear: "RelA+ expression results in elevated levels of ppGpp, which are then hydrolyzed by SpoTH."

Feedforward growth rate control mitigates gene activation burden

Response to reviewers

We highly appreciate the reviewers' comments on our manuscript. Our responses can be found below, with the reviewers' comments in black, and our responses in blue. We hope the reviewers are satisfied with our responses, and thank them again for helping us improve the quality of this manuscript.

Response to Reviewer 1:

Major comments:

1. The trade-off between the nominal growth rate and the relative growth rate and the non-monotonic curves shown in Fig.1b is very interesting. However, in the main text, this is not explained in detail. Is this related to the burden caused by the controller itself? Is it because of the saturation level of ppGpp? I believe that this is important information to have in the main text so the reader will understand under what conditions the controller works best intuitively.

Response:

We thank the reviewer for this comment. Yes, the non-monotonicity occurs because of the load SpoTH expression places on cellular resources. Once a sufficient amount of ppGpp has been removed through SpoTH expression, the burden effects from further SpoTH expression overwhelm the upregulation in growth rate due to the removal of ppGpp. This is discussed in supplementary notes 2 and 3. To make this clearer in the main text, we have added to Figure 1-a a dashed flat headed arrow from SpoTH to ribosomes representing the load SpoTH expression places on ribosomes. Furthermore, we have changed the style of the plot in Figure 1-b and have used a bar chart. This also allows easier comparison with the experimental data of Figure 2-c (as suggested by the reviewers minor comment 3).

The non-monotonicity is discussed in lines 66-71 and 78-82. We have added/modified our commentary in the main text discussing the trade-off between nominal growth rate and the relative amount by which growth rate can be actuated (lines 92-102). Then, in line 122 we discuss the implications of this trade-off on the controller performance. Starting in line 231, we discuss the implications of this trade-off when using the controller to make protein production rates robust to the GOI activation.

2. There seems to be inconsistency between the well-maintained growth rate and a strong decrease in the GFP production rate in Fig. 9 and 13. I understand the authors have some discussion on it. But I suggest the author move the discussion into the main text. I believe this is reasonable if we consider that the resource is also very limited in the system with the controller even if the growth rate is maintained very well.

Response:

We thank the reviewer for this suggestion. The results of Fig. 9 and Fig. 13 are expected based on how differently growth rate and GFP production (proxy for amount of free gene expression resources in the cell) respond to SpoTH expression (Compare Figure 2 to Figure 7). Both (GFP production rate and growth rate) response curves have a similar qualitative behavior where the corresponding quantity initially increases with SpoTH expression, peaks, and then

begins to decrease with SpoTH expression. However, we observe that the GFP production rate peaks at a lower SpoTH expression level than growth rate (Compare Figure 2 to Figure 7). This occurs because as SpoTH is expressed, growth rate varies due to two factors: changes in cellular resources used to produce growth proteins and direct regulation of growth proteins by ppGpp [1, 2]. Hence, even if the free cellular resource concentration is decreasing with SpoTH expression, the decrease in ppGpp can compensate for this through direct regulation of growth proteins and can overall increase growth rate (Equation (30) in Supplementary note 3 makes this mathematically precise). This implies that for the feedforward controller the SpoTH RBS that keeps growth rate constant is one where the GFP production rate is decreasing with GOI expression (equation (31) makes this mathematically precise). However, we want to emphasize that from Figure 9 and 13, even though GFP production rate decreases with GOI activation, the CL system with the RBS that keeps growth rate constant, still outperforms the OL system in the sense that it yields higher GFP production rate for a fixed GOI activation.

While we agree that this lack of correlation between how growth rate and GFP production rate respond to SpoTH expression provides insight on the inner-working of the controller (and provides confidence in our model), we have opted to move all content pertaining to GFP production to the discussion section (lines 217-250). The rationale for making this decision is that we want to streamline the paper to solely focus on growth rate control (we even changed the title of the paper to reflect this) and we believe that the discussion on GFP production rate distracts from this central message. We further dedicate a large portion of the discussion to discuss this and its relevance to the key issue in synthetic biology of keeping protein expression rates constant as a GOI is activated [3]. This decision was also motivated based on Reviewers 2 Major Comment 2.

Minor comments:

1. The authors showed the design of RelA+ in Fig. 1a but did not have any description of it. I suggest that this is either be removed from the diagram and left to Fig. 4 or added some description of the design idea of RelA+ in the beginning.

Response:

We thank the reviewer for bringing this point to our attention. We have added a discussion of RelA+ in the introduction starting in line 45 and in the caption of figure 1.

2. The authors need to explain the direct link from SpoTH to Ribosome in Fig. 2b, in the caption as well as in the main text. Maybe it is a good idea to relate it to the trade-off in the first subsection of Results.

Response:

We thank the reviewer for this suggestions. We have added a description of this interaction in the caption of Figures 1, 2, and 4 indicating that this direct link comes from the fact that SpoTH expression places a load on cellular resources, particularly ribosomes, since they are sequestered for its expression. We discuss the implications of this interaction, the non-monotonic behavior, and the trade-off between nominal growth rate and relative growth rate actuation as SpoTH is expressed in lines 62, 92, 122, and 231. In summary, this interaction is the root cause of the non-monotonic behavior the reviewer pointed in the first major comment and also implies that the maximum achievable growth rate via SpoTH expression will be lower than that with a cell with no ppGpp and no SpoTH expression (MG1655 in Figure 1-b).

3. It is kind of difficult to map Fig. 2c with Fig. 1b.

Response:

We thank the reviewer for pointing this out. This along with the reviewer's first major comment motivated us to change Figure 1-b to a bar graph, which is easier to map to figure 2-b.

4. Given that the nominal growth rate in OL and CL systems is almost the same, is it necessary to normalize the growth rate in Fig. 3 and 5, especially in Fig. 5, where the growth rate matches very well?

Response:

We thank the reviewer for their comment. We have decided to keep the growth rates normalized because it makes it easier to assess the percentage-wise variations in growth rate, especially across different nominal growth rates (Fig. 3/5 panels e-g). Also, the insets have the same scales across all nominal growth rates and thus they emphasize the differences in growth rate between the three nominal conditions while also allowing us to vary the scale of the normalized plot to see variations as the GOI is expressed.

5. In the system with RelA+, should the inhibition of Ribosomes mediated by RelA+ also be considered as well? Is there any potential trade-off as well, similar to the SpoTH?

Response:

We thank the reviewer for their question. It is correct, RelA+ expression requires free ribosomes and hence there is an inhibition from RelA+ to ribosomes, similar to that of SpoTH. However, different from the situation with SpoTH, this inhibition does not lead to a non-monotonic response between growth rate and RelA+ expression (Fig. 4-c). This occurs because the intended interaction (via ppGpp synthesis) and the unintended interaction (via ribosome sequestration) are both inhibitions. RelA+ expression also does not change the qualitative behavior of growth rate vs SpoTH (Fig. 4-d) and that of growth rate vs GOI expression (Fig. 5) in the CL system. The only trade-off arising from the load RelA+ places on free cellular resources, is that the maximum growth rate achievable as SpoTH is expressed is lower than when there is no RelA+ expression. However, notice that this burden effect on growth rate is minimal by comparing Fig. 4-c (no SAL induction) with Fig. 4-d (max aTc induction). For these reasons, we have decided not to include this hidden interaction in the figure diagrams since it will clutter the circuit diagrams without providing useful insight.

6. Two relevant references (PMID: 33558556, 32251409) are worth to be cited if possible when discussing the effects of decreased growth rate on the performance of engineered cells (page 2, line 25).

Response:

We thank the reviewer for providing these references. Although 33558556 was already included as a reference where the reviewer suggested (Ref number 17 in the previous version of the manuscript), we are thankful that they brought 32251409 to our attention. We have gone ahead and included it since this reference demonstrates the key issue of having a variable growth rate on genetic circuit dynamics.

7. What is the potential reason for the increase in the growth rate under RFP production in Fig. 8 and 12?

Response:

We thank the reviewer for this question. Recall that for Fig. 8 and 12 SpoTH and RFP are expressed on the same mRNA. Therefore, by tuning the SpoTH RBS, we control the expression of SpoTH relative to RFP. A stronger SpoTH RBS, implies a higher SpoTH production rate for a fixed RFP production rate. Hence, for this strong RBS we make SpoTH at a level that over compensates the decrease in growth rate caused by RFP expression. This is captured by our mathematical model (part of Supplemental note 2) and shown in Fig. 1 (which is included in Supplemental note 2).

Response to Reviewer 2:

Major comments:

1. The general idea of reducing cell growth by producing a toxin and reverting the effect by expressing an antidot is not new. This is the underlying principle of the toxin-antitoxin systems,

Figure 1: **Feedforward controller as the SpoTH RBS strength is varied.** A proxy for RFP expression (\bar{c}_y) vs a proxy for growth rate (\bar{z}). The orange curve corresponds to the open loop system (OL) where RFP (y) is expressed in isolation. The blue curve corresponds to the closed loop system (CL) where SpoTH and RFP are expressed under the same mRNA. The different blue curves correspond to different values of the SpoTH RBS strength (γ). When the SpoTH RBS is weak, the growth rates of the OL and the CL systems almost coincide since only a small amount of SpoTH is produced as RFP is activated. For a strong RBS, the growth rate initially increases as RFP is expressed since the positive growth rate actuation effects of expressing SpoTH overwhelm the burden on cellular resources from expressing RFP.

of coexpressing the antibiotic biosynthetic operons and resistance proteins, of the secondary metabolites and the relevant efflux pumps, and many others. A possible advantage of the SpoTH/RelA+ is a possibility of fine-tuning the growth rate of the cells, even though it is unclear whether SpoTH/RelA+ system is any better in this sense than any other of the aforementioned systems.

Response:

We thank the reviewer for raising this point. We now note this general similarity of approach with toxin/antitoxin systems in the discussion (starting at line 276). We specifically note the analogy between SpoTH and the anti-toxin and RelA+ and the toxin, but we also point out significant differences (detailed below here). In general, a toxin-antitoxin system may serve as an alternative approach to using SpoTH/RelA+. Although, it appears that some toxin-antitoxin systems exploit the growth control properties of ppGpp and hence there may be some overlap with our approach [4]. However, there are important advantages of using RelA+ and SpoTH directly as control knobs, as detailed below.

The key advantage of using the growth control knobs (RelA and SpoT) of the ppGpp pathway is that these have been studied extensively by microbiologist since 1969 [5]. A plethora of tools have been developed to study this system, which can now be exploited in an engineering context, such as we have done in this study. For example, our usage of the library of strains with varying ppGpp levels (CF94x) allowed us to vary ppGpp and nominal growth rate before we introduced RelA+ [6]. For the experiments where we used these strains, our controller only needed to express SpoTH and not RelA+, thus reducing the number of genetic modules that make up our controller and thus also allowing us to test the basic feedforward control design in a simpler context (thus one potential advantage to toxin-antitoxin systems). In addition to this, the ppGpp literature provided us with the design of SpoTH and RelA+, both of which are well-characterized (we discuss this characterization further below when addressing the reviewers Major Comment 2) [7, 8]. The ppGpp community is still very active and developing new tools. To this end, we discuss (line 283) recently developed enzymes that serve as alternatives to SpoTH such as Mesh1 and SpoT E319Q [11, 10, 12]. Therefore, by framing the controller as part of the ppGpp system, we unlock the potential to use all of these tools and to readily integrate any future advances. Additionally, given that ppGpp is the major

source of growth rate control in exponential growth and that SpoT is responsible for setting the basal ppGpp level in the cell and growth rate, our approach works in synergy with and leverages the natural growth rate control circuitry in the cell [13, 14].

Finally, given that ppGpp and the RelA/SpoT homologs (RSHs) are universally conserved in bacteria and even appear in eukaryotes, including in humans, we believe that our controller can be easily transported across organisms [15]. The generalization of our controller to other organisms can play a key role when designing multi-organism systems where each member is endowed with growth rates that are robust to gene actuation. We discuss this in line 266.

2. The authors presume that expression of SpoTH or RelA+ affects cell growth (one way or the other) by changing the intracellular level of ppGpp. Although this is generally a reasonable assumption, without direct measurements of the ppGpp level, it is nothing but an assumption. Thus, following the authors logic (see below), decreased growth of RelA+ expressing cells could be due to the ‘ribosome sequestration’ rather than ppGpp accumulation. Therefore, either the levels of ppGpp levels need to be measured directly or any relation to these levels need to be presented as only tentative.

Response:

We thank the reviewer for these comments. In short, data directly showing that SpoTH and RelA+ change ppGpp level exist already and that is exactly what we use in this paper [7, 8, 9, 10]. Specifically, ppGpp concentration over time when SpoTH is expressed was measured in [7] as shown in Fig. 2-a from reference [7] and reported down here for extra clarity. The data demonstrates that SpoTH expression causes a decrease in ppGpp concentration. Similarly, in [8, 9, 10], ppGpp concentration over time was measured when RelA+ was expressed at different levels as shown in Fig. 2-b (top) from reference [8] and reported here for completeness. The data demonstrates that RelA+ expression increases ppGpp concentration.

Additional evidence that RelA+ expression decreases growth rate due to an increase in ppGpp rather than free ribosome/resource sequestration lies in figure 2-c and figure 4-c,d. Specifically, we observe that SpoTH expression cannot upregulate growth rate in MG1655 (figure 2-c), which can be attributed to the fact that there is not enough basal ppGpp (see our comments in line 92). Therefore, if the decrease in growth rate in MG1655 as RelA+ is expressed (figure 4-c) were due to ribosome sequestration and not to an increase in ppGpp level, then SpoTH expression would only burden the cell and cause a decrease in growth rate (MG1655 in figure 2-c). However, when RelA+ is expressed in MG1655, we observe an increase in growth rate as SpoTH is expressed (figure 4-d), thus indicating that RelA+ expression elevates ppGpp levels. Additionally, the fact that SpoTH expression increases the growth rate in MG1655 close to the level with no RelA+ and no SpoTH expression implies that the burden that these molecules placed on growth rate is small relative to the amount by which growth rate changes due to changes in ppGpp from RelA+/SpoTH expression.

In line 62 and line 139, we modified the main text to explicitly state that ppGpp levels were previously measured as these two enzymes are expressed.

3. ll. 95-96. The authors claim that activation of rfp transcription from a plasmid negatively affects growth rate because it “sequesters ribosomes”. The same reason is used for explaining why expression of other proteins of interest reduces cell growth. However, expression of many proteins can negatively affect cell for a number of reasons other than ribosome sequestration, while expression of the others may have no effect. Accordingly, what is the evidence that slow cell growth is caused by sequestration of the ribosomes? How large is the fraction of ribosomes that would be ‘sequestered’ by translating a single gene

Response:

We thank the reviewer for this question. In short, the fact that RFP and other unneeded proteins’ overexpression causes a sequestration of cellular resources required for translation and of ribosomes in particular was experimentally demonstrated in prior work [16, 17, 18]. The fact that overexpression of unneeded proteins causes a decrease in growth rate that is correlated

a)

Fig. 2. Rate of ppGpp decay in strains carrying various *spoT* deletions as a sole source of *spoT*-dependent ppGppase. Cultures were labelled with ^{32}P as described in the *Experimental procedures*, and starved for isoleucine by the addition of valine to $500\ \mu\text{g ml}^{-1}$. Fifteen minutes after valine addition, isoleucine was added to $500\ \mu\text{g ml}^{-1}$ and samples were taken at 0.5 min intervals. The graph shows counts of ppGpp relative to the time of addition of isoleucine. Deletion mutants tested are: $\Delta 407-702$ (circles), $\Delta 376-702$ (squares), $\Delta 204-466$ (triangles), and $\Delta 235-702$ (diamonds).

b)

FIG. 2. Changes in ppGpp level after induction of pSM10 and pSM11. Cells containing pSM11 (panel A) or pSM10 (panel B) were grown and uniformly labeled with ^{32}P in glucose minimal medium supplemented with most amino acids to an A_{600} of 0.2, when different concentrations of IPTG were added. Samples were withdrawn at times indicated and assayed for their ppGpp content. The induction of pSM12 with IPTG had no effect on ppGpp levels (data not shown). ■, no IPTG; □, 20 μM IPTG; Δ , 50 μM IPTG; \circ , 200 μM IPTG.

Figure 2: **ppGpp concentration as SpoTH and RelA+ are expressed.** (a) The ppGpp concentration over time when several mutants of SpoT are induced, including SpoTH as denoted by a red arrow. The data was extracted from [7]. (b) The ppGpp concentration over time when RelA+ is expressed at several levels (top plot). Increasing RelA+ concentration is denoted by a purple arrow. pSM11 corresponds to a plasmid expressing RelA+ and pSM10 corresponds to a plasmid expressing RelA. The data was extracted from [8].

with a general decrease in the expression rate of other genes was also experimentally demonstrated before [19]. However, we agree with the reviewer that while ribosome sequestration may contribute to the decrease in growth rate as RFP is expressed, it is not necessarily the only factor. We therefore clarified this point by rephrasing some of our former statements that may have created confusion in this regard. Specifically, in the results section (line 104) and discussion (line 209), we now state that “RFP induction sequesters cellular resources, including ribosomes, and thus negatively affects growth rate”. However, we emphasize that the causation of the growth rate defects is unimportant since the controller is independent of this causality. What matters for our controller to achieve constant growth rate as a GOI (RFP) is activated is: (a) RFP and SpoTH are co-expressed and (b) SpoTH expression increases growth rate (Figure 2 in the main text). Altogether, these facts allow us to engineer the controller to keep growth rate constant as the GOI is activated. We have changed the title of the paper to remove any confusion and stress that our controller is a growth rate controller, thereby removing any unnecessary reference to the causality between ribosome sequestration and growth defects.

4. ll. 73-75 and ll. 784-797. Measuring ribosome concentration by following GFP expression is a questionable approach. It is based on yet another unsubstantiated assumption that translation elongation rate is not affected by varying ppGpp concentrations. However, given that ppGpp controls transcription of not only rRNA but also tRNA as well as expression of amino acid biosynthetic operons and influx pumps, there are all the reasons to expect that elongation rate would be different in cells with high or low levels of ppGpp. Therefore, without a more direct measurement of ribosome content, it is hard to know how SpoTH expression affects the

amount of free ribosomes in the cell.

Response:

We thank the reviewer for the comment, with which we completely agree. Changes in ppGpp (through SpoTH expression) will change the concentration of a few cellular resources responsible for protein production, not just ribosomes. In this study, GFP is mainly an additional metric of interest when designing genetic circuits, that is, the effect of the expression of one gene on any other gene [18, 19]. We use it to highlight the important fact that keeping growth rate constant as RFP is expressed does not necessarily keep production rate of other genes constant and vice-versa. In order to remove any confusion, we have moved all material pertaining to the GFP monitor to the discussion (line 217-251). Secondly, we have modified our statements to say that we use the GFP monitor to investigate how SpoTH expression/the controller affect protein production rates rather than ribosomes level.

Minor comments:

1. 45-46: It is confusing that “to achieve derepression of the rRNA” one activates “ppGpp synthesis activity”. The next sentence is even more confusing, not only grammatically, but also its sense is unclear: “RelA+ expression results in elevated levels of ppGpp, which are then hydrolyzed by SpoTH.”

Response:

We thank the reviewer for the feedback. We have gone ahead and re-written the paragraph entirely.

References

- [1] B. Wang, P. Dai, D. Ding, A. D. Rosario, R. A. Grant, B. L. Pentelute, and M. T. Laub, “Affinity-based capture and identification of protein effectors of the growth regulator ppgpp,” *Nature Chemical Biology*, vol. 15, pp. 141–150, 2 2019.
- [2] B. Wang, R. A. Grant, and M. T. Laub, “ppgpp coordinates nucleotide and amino-acid synthesis in e. coli during starvation,” *Molecular Cell*, vol. 80, pp. 29–42.e10, 10 2020.
- [3] H. H. Huang, Y. Qian, and D. Del Vecchio, “A quasi-integral controller for adaptation of genetic modules to variable ribosome demand,” *Nat. Commun.*, vol. 9, no. 1, pp. 1–12, 2018.
- [4] S. Jimmy, C. Kumar Saha, T. Kurata, C. Stavropoulos, S. Raquel Alves Oliveira, A. Koh, A. Cepauskas, H. Takada, D. Rejman, T. Tenson, H. Strahl, A. Garcia-Pino, V. Haurlyliuk, and G. C. Atkinson, “A widespread toxin antitoxin system exploiting growth control via alarmone signaling,” vol. 117, no. 19, 2020.
- [5] M. Cashel and J. Gallant, “Two compounds implicated in the function of the RC gene of escherichia coli,” *Nature*, 1969.
- [6] E. Sarubbi, K. E. Rudd, and M. Cashel, “Basal ppGpp level adjustment shown by new spoT mutants affect steady state growth rates and rrnA ribosomal promoter regulation in Escherichia coli,” *MGG Mol. Gen. Genet.*, vol. 213, no. 2-3, pp. 214–222, 1988.
- [7] D. R. Gentry and M. Cashel, “Mutational analysis of the Escherichia coli spoT gene identifies distinct but overlapping regions involved in ppGpp synthesis and degradation,” *Mol. Microbiol.*, vol. 19, no. 6, pp. 1373–1384, 1996.
- [8] G. Schreiber, S. Metzger, E. Aizenman, S. Roza, M. Cashel, and G. Glaser, “Overexpression of the relA gene in Escherichia coli,” *J. Biol. Chem.*, vol. 266, pp. 3760–3767, feb 1991.

- [9] A. Svitil, M. Cashel, and J. Zyskind, “Guanosine tetraphosphate inhibits protein synthesis in vivo. A possible protective mechanism for starvation stress in *Escherichia coli*,” *J. Biol. Chem.*, vol. 268, pp. 2307–2311, feb 1993.
- [10] M. Zhu and X. Dai, “Growth suppression by altered (p)ppGpp levels results from non-optimal resource allocation in *Escherichia coli*,” *Nucleic Acids Res.*, vol. 47, no. 9, pp. 4684–4693, 2019.
- [11] D. Sun, G. Lee, J. H. Lee, H. Y. Kim, H. W. Rhee, S. Y. Park, K. J. Kim, Y. Kim, B. Y. Kim, J. I. Hong, C. Park, H. E. Choy, J. H. Kim, Y. H. Jeon, and J. Chung, “A metazoan ortholog of SpoT hydrolyzes ppGpp and functions in starvation responses,” *Nat. Struct. Mol. Biol.*, vol. 17, no. 10, pp. 1188–1194, 2010.
- [12] R. Harinarayanan, H. Murphy, and M. Cashel, “Synthetic growth phenotypes of *Escherichia coli* lacking ppGpp and transketolase A (tktA) are due to ppGpp-mediated transcriptional regulation of tktB,” *Mol. Microbiol.*, vol. 69, pp. 882–894, aug 2008.
- [13] J. Ryals, R. Little, and H. Bremer, “Control of rRNA and tRNA syntheses in *Escherichia coli* by guanosine tetraphosphate,” *J. Bacteriol.*, vol. 151, no. 3, pp. 1261–1268, 1982.
- [14] K. Potrykus, H. Murphy, N. Philippe, and M. Cashel, “ppGpp is the major source of growth rate control in *E. coli*,” *Environ. Microbiol.*, vol. 13, no. 3, pp. 563–575, 2011.
- [15] D. Ito, H. Kawamura, A. Oikawa, Y. Ihara, T. Shibata, N. Nakamura, T. Asano, S.-I. Kawabata, T. Suzuki, and S. Masuda, “ppGpp functions as an alarmone in metazoa,” *Commun. Biol.*, vol. 3, p. 671, dec 2020.
- [16] J. Vind, M. A. Sørensen, M. D. Rasmussen, and S. Pedersen, “Synthesis of proteins in *Escherichia coli* is limited by the concentration of free ribosomes. Expression from reporter genes does not always reflect functional mRNA levels,” 1993.
- [17] S. Klumpp, Z. Zhang, and T. Hwa, “Growth Rate-Dependent Global Effects on Gene Expression in Bacteria,” *Cell*, vol. 139, no. 7, pp. 1366–1375, 2009.
- [18] A. Gyorgy, J. I. Jiménez, J. Yazbek, H. H. Huang, H. Chung, R. Weiss, and D. Del Vecchio, “Isocost Lines Describe the Cellular Economy of Genetic Circuits,” *Biophys. J.*, vol. 109, no. 3, pp. 639–646, 2015.
- [19] F. Ceroni, R. Algar, G.-B. Stan, and T. Ellis, “Quantifying cellular capacity identifies gene expression designs with reduced burden,” *Nat. Methods*, vol. 12, no. 5, pp. 415–418, 2015.

Reviewers' Comments:

Reviewer #1:

Remarks to the Author:

Thank the authors for the comprehensive and thoughtful responses. All my comments are adequately addressed. I am supportive of the publication of this work in Nature Communications.

Reviewer #2:

Remarks to the Author:

The authors have adequately addressed my concerns.

Feedforward growth rate control mitigates gene activation burden

Response to reviewers

We highly appreciate the reviewers' comments on our manuscript. Our responses can be found below, with the reviewers' comments in black, and our responses in blue. We hope the reviewers are satisfied with our responses, and thank them again for helping us improve the quality of this manuscript.

Response to Reviewer 1:

1. Thank the authors for the comprehensive and thoughtful responses. All my comments are adequately addressed. I am supportive of the publication of this work in Nature Communications.

Response:

We thank the author for their time in reviewing the article. Their input and feedback has greatly improved the quality of the manuscript.

Response to Reviewer 2:

Major comments:

1. The authors have adequately addressed my concerns.

Response:

We thank the author for their time in reviewing the article. Their input and feedback has greatly improved the quality of the manuscript.